# Animals Like Us: Leveraging the Negativity Bias in Anthropomorphism to Reduce Beef Consumption

**DOI:** 10.3390/foods10092147

**Published:** 2021-09-10

**Authors:** Ziad Choueiki, Maggie Geuens, Iris Vermeir

**Affiliations:** BE4LIFE, Department of Marketing, Innovation and Organisation, Faculty of Economics and Business Administration, Ghent University, 9000 Ghent, Belgium; Maggie.Geuens@UGent.be (M.G.); Iris.Vermeir@UGent.be (I.V.)

**Keywords:** meat reduction, anthropomorphism, negativity bias

## Abstract

Our current work contributes to the literature of meat consumption reduction. Capitalizing on the inherent humanizing characteristic of anthropomorphism coupled with leveraging negativity bias, we created a novel approach to reduce meat-eating intention. Using on-pack product stickers, we compare an anthropomorphic message stressing the capacity to experience pain with two other anthropomorphic messages that have been used before in the literature (intelligence and pro-social behavior of animals). We find that an on-pack pain anthropomorphic sticker reduces purchase intentions of the meat product and intention to consume meat in general and is more effective than stickers displaying pro-social or intelligence messages. We also show that the pain message’s negative impact on purchase intention is serially mediated by anticipatory guilt and attitude towards meat. In addition, we show that the differential effectiveness of the anthropomorphic messages can be explained by the negativity bias. That is, when the pro-social and intelligence messages were formulated in a negative way (as is pain), all three messages were equally effective at reducing intention to purchase meat and increase intention to reduce meat consumption.

## 1. Introduction

For the majority of history, the place of meat in humans’ diet has remained one of an honored and celebrated food item for both its perceived health benefits and its influence on societies’ cultural, pro-social and economic development [1]. Not surprisingly this has led us to consider eating meat as natural, normal, necessary, and nice [2]. It is only recently that meat came under intense scrutiny which increasingly highlighted the detrimental impact of meat production and consumption on animal welfare, human health and the environment [3,4,5,6,7,8,9]. However, despite this scientific evidence, meat demand is expected to further increase in the coming years [10,11,12,13], putting intense strain on our already exhausted ecosystems. Hence, the necessity to seek strategies to counter this increasing demand and to reduce consumers’ intentions to purchase meat.

In order to arrive at successful intervention strategies, it is important to realize that meat consumption often elicits a form of cognitive dissonance [14] in meat-eaters, referred to as the “meat paradox” [15]. A quite tangible state of discomfort, the meat paradox is generated by a disparity between one’s beliefs and one’s actions. For a meat-eater, this could be presented as the fact that he or she loves to eat meat (i.e., the act) and that the meat-eater sees him/herself as an ethical human being that does not wish to harm or kill animals involved in his/her meat-eating habit (i.e., the belief). Cognitive dissonance is an unpleasant state which people seek to avoid or reduce [14,16]. In the context of meat consumption, several researchers [16,17] have already documented that consumers either employ prevention mechanisms aimed at blocking this state of dissonance from arising or use perceptual strategies to reduce the dissonance and reinforce the strong held beliefs regarding their meat eating habits. 

As such, effective strategies to reduce consumers’ meat-eating intentions should try to counter consumers’ prevention mechanisms or perceptual strategies leaving the cognitive dissonance intact and thus leading the consumer to change his or her behavior rather than, for example, avoiding or changing the informational value of the intervention strategy. With respect to eating meat, an often used perceptual strategy is “Denial of Animal Mind”, meaning that people can easily justify harming or eating animals they believe are unable to think, feel or suffer [16,18,19,20]. An ideal intervention strategy to counter consumers’ denial of animal mind seems the use of anthropomorphism. Attributing human characteristics to animals makes it much harder for consumers to deny animal mind. Initial research already found promising results in this respect [21]. However, this research focused on pro-social behavior only and ignored the ability to suffer or experience pain. 

The purpose of this study is to close this gap in the literature and to compare different types of anthropomorphism in their ability to counter consumers’ perceptual strategies and thereby reducing meat consumption intentions. We propose that focusing on this aspect of animal mind might be even more promising because of the negativity bias. That is the innate tendency in humans to give greater weight to negative entities (events, objects and personal traits) than they do to positive ones [22,23]. 

### 1.1. Denial of Animal Mind

A commonly employed perceptual strategy is the “denial of animal mind”. A form of neutralization, the act of denying the mind of the victim reduces the moral indignation at the self for committing any transgression against said victim. Moral judgements towards the self diminish if the target is seen as unable to think, feel or suffer [24,25,26] as such reducing or eliminating the state of cognitive dissonance arising from the maltreatment or abuse of the victim. In support of the “denial of animal mind” strategy, Bastian et al. [20] showed that animals viewed as appropriate for human consumption were attributed with lower mental capacities such as agency-related capacities (self-control, morality and memory) and experience-related capacities (hunger, fear and pleasure), with this perception being a dissonance reducing mechanism employed by the respondents. Similar results are observed by Ang et al. [27] where meat-eaters accredited less mental capacities to meat animals in comparison to pet animals, furthermore those same meat-eaters were less likely to consider the killing of animals for food as morally wrong. Another study by Loughnan et al. [28] reached similar conclusions where the eating of meat reduced the moral standing of animals and indirectly reduced the attribution of mental states necessary to experience pain. Likewise, people who consume greater amounts of meat tend to believe that animals do not experience pain in the same way as humans do [29,30]. The denial of animal mind as a strategy to reduce meat-related cognitive dissonance is best explained by Bastian et al. [20] where they state that the adoption of this approach enables meat-eaters to reduce their concern for the animals’ welfare, justifies their harm in the course of meat production and facilitates meat consumption by trivializing the ordeal of meat animals in such circumstances. 

### 1.2. Anthropomorphism as Solution

A proposed solution to combat this perceptual “denial of animal mind” strategy is the use of Anthropomorphism which can be defined as “*the tendency to imbue the real or imagined behavior of non-human agents with humanlike characteristics, motivations, intentions, or emotions*” (Epley et al. [31], p. 864). Prior studies showed that anthropomorphism can successfully change people’s behavior [32,33,34]. In their 2018 paper, De Bondt et al. [35] identified how anthropomorphized packaging can result in a positive product evaluation by eliciting aesthetic appeal. In a similar fashion, Cooremans and Geuens [36] successfully reduced the aversion of misshapen produce by adding anthropomorphic traits (i.e., smiling faces) which led to enhanced taste perceptions, an increase in purchase intentions and product choice of what would usually be a quite unappealing and aesthetically poor-looking food product. 

In the context of meat consumption, attributing humanlike traits to animals might have the effect of undoing the denial of animal mind [16]. Chan [37] states that a successful usage of anthropomorphism in protection efforts of an animal depends on three specific traits: the animal’s cognitive ability (“intelligence”), the animal’s pro-social behavior (“pro-social”) and the animal’s ability to suffer or experience pain (“pain”). On a similar note, Piazza et al. [38] commented that people’s moral concern for animals is based partially on how much the animal possesses “mind”, that is capacity to suffer, experience pleasure and possesses intelligence. Also Leach et al. [39] revealed that a broader range of animal traits like the ability to feel love, to share food with others and to look for deceased family members raised moral beliefs regarding the animals and by extension increased wrongness and guilt associated with their consumption.

In this context, past studies started to explore the effectiveness of anthropomorphism by attributing animals either cognitive ability or pro-social behavior, but these studies yielded divergent results. Concerning intelligence anthropomorphism, Piazza and Loughnan [18] revealed that, even though the intelligence of an animal is a key factor in determining its moral status (animals with high moral standing are perceived as deserving moral concern and should not be harmed), this information is disregarded when the animal is used as food in the participant’s culture and the respondents’ own consumption of the animal’s meat is at stake. Indeed, investigating three different animals, attributed intelligence only mattered for tapirs and a fictional animal race, but not for the animal that was included in the respondents’ diets, namely pigs. In contrast, focusing on pro-social behavior anthropomorphism, Wang and Basso [21] depicted pigs in animal-animal friendships and human-animal friendships (labeled as “the friendship metaphor”) which did lead to negative attitudes towards pork, as well as reduced intentions to purchase pork products. Furthermore, they revealed that increased levels of anticipatory guilt mediated the relationship between anthropomorphism on consumers’ attitudes and purchase intentions. However, as is the case for intelligence anthropomorphism, also pro-social behavior anthropomorphism does not seem to be universally effective. Indeed, when attempting to replicate for cows, the anthropomorphic friendship-metaphor did not yield any significant results. The authors explained this by an incongruent association between the anthropomorphic friendship message and the metaphoric association of cows, anger and irritability in English or the fact that cows are usually portrayed as somewhat idiotic [21]. The foregoing studies indicate that anthropomorphism shows potential, but that results depend on the type of animal it is applied to, but also on the specific anthropomorphic trait that is used. So far, the third trait (i.e., “pain”) reflected in animal mind, that is attributing animals the ability to suffer or to experience pain, has been ignored. This is surprising as its potential may actually surpass the other traits (intelligence and pro-social behavior) used to counteract denial of animal mind. We aim to add to the literature by also investigating the capacity to suffer or experience pain as an anthropomorphic aspect and hence to include the full range of anthropomorphic traits related to animal mind (i.e., intelligence, pro-social behavior and pain) in one and the same study.

### 1.3. Pain Anthropomorphism and the Negativity Bias

Previous research focused on the animal’s cognitive ability and the animal’s pro-social behavior as anthropomorphic traits to decrease meat consumption. These anthropomorphic traits are usually formulated in a positive way (e.g., “love to play lots of interactive games” [21]; “the superior cognitive abilities” [38]). We argue that respondents’ motivated cognition to justify their meat-eating behavior will be greatly impacted when a negative aspect like suffering or experiencing pain is paired with meat consumption. Drawing from the theory of “negativity bias”, we capitalize on the inherent tendency of humans to give greater weight to negative than positive entities (simply put “bad is stronger than good”) [23]. Specifically, we consider that a mixture of enjoying meat and the suffering of animals can be a prime example of “negativity dominance” (a subclass of negativity bias) where a combination of positive (i.e., meat consumption) and negative (i.e., pain and suffering of animals) entities will yield an end evaluation that is more negative than the algebraic sum of the subjective values of those individual entities [22]. 

### 1.4. Research Aims and Hypotheses

Successful previous meat reduction interventions intended to curb meat consumption employed a variety of methods that ranged from a focus on animal welfare, human health, environmental concerns or a mix to motivate respondents to change their behavior [40]. The majority of these interventions solely used text with no images being involved [41,42,43,44,45,46]. Of those studies in which animal welfare was utilized as a meat reducing method, the fashion in which it was communicated was in a sterile and non-anthropomorphic way (i.e., “…Maltreatment, such as the castration of male piglets without anesthesia … is also still common” [45]). Considering the above, we set out to test whether anthropomorphizing animals stressing the animal’s cognitive ability (“intelligence message”), the animal’s pro-social behavior (“pro-social message”) or the animal’s ability to suffer (“pain message”) reduces meat consumption intentions. 

**Hypothesis** **1** **(H1).**
*On-pack stickers using anthropomorphic messages (intelligence, pro-social or pain) (vs. no sticker or just-cow sticker) induce (a) lower meat purchase intentions and (b) enhanced intentions to reduce general meat consumption.*


As elucidated by the negativity bias theory, humans have a tendency to give a greater weight to negative entities [22] and respond more strongly to negative versus positive stimuli [47,48,49]. Coupled with the fact that intelligence and sociability in animals are usually portrayed in a positive light [18,21], we predicted that the pain message would be more effective than the pro-social and intelligence message in reducing meat purchase intentions and increase intentions to reduce meat consumption because of its inherent negativity that allows it to tap into the power of the negativity bias. As such we hypothesize:

**Hypothesis** **2** **(H2).**
*The pain message is more effective in (a) lowering purchase intentions and (b) increasing intentions to reduce meat consumption compared to the other two messages (intelligence and pro-social).*


According to the theory of planned behavior [1], behavioral intention is shaped by three key antecedents: attitude (i.e., favorable or unfavorable evaluation of the behavior in question), subjective norms (i.e., importance of others’ opinions) and perceived behavioral control (the perceived control of a behavior toward a certain action). As a general rule, an individual’s intention to perform a certain behavior is positively correlated with these three elements. In the context of our study, we test anticipatory guilt and attitudes towards meat as possible serial mediators of animal pain anthropomorphism on purchase intentions, similar to the work of Wang and Basso [21]. The consumption of meat carries with it the risk of experiencing guilt and shame due to the unethical nature in which meat is procured [22] and by extension consumed [19]. This could motivate individuals to deny the mind of their victims to reduce the guilt they experience from their actions. Using anthropomorphic messages could counter-act the guilt reducing mechanism of denial of animal mind since these messages showcase the same capacities that meat-eaters are trying to refute (i.e., pro-social behavior, intelligence and capacity to experience pain). Leach et al. [31] already found that associating human-like characteristics with animals increases guilt feelings about eating meat. Building on the above, we expect the effect of anthropomorphism to occur because attributing a human characteristic to animals will lead to greater guilt (in comparison to no anthropomorphism) which in turn will lead to less positive attitudes towards the meat and finally lowered purchase intentions as predicted by the theory of planned behavior. As such, we hypothesize that:

**Hypothesis** **3** **(H3).**
*The effect of anthropomorphic messages on purchase intentions is mediated via anticipatory guilt and attitudes towards meat.*


Finally, we argue that if the inherent negativity of the pain message explains its more effective impact on reducing meat consumption intentions compared to addressing intelligence and pro-social anthropomorphic traits, then framing intelligence and pro-social traits in a negative way would equalize the impact of the three anthropomorphic traits. It can be stated that depictions of pain are inherently negative while pro-social and intelligence capacities are mainly portrayed in a positive light. The lack of negativity in these two anthropomorphic states, in comparison to pain, might explain the inconsistency of anthropomorphism’s effect on denial of animal mind and the success of meat-eaters’ motivated cognition in by-passing the humanizing effect of anthropomorphism. As such, we hypothesize that:

**Hypothesis** **4** **(H4).**
*On-pack stickers using anthropomorphic messages in the form of negatively oriented intelligence and pro-social traits are as effective as on-pack stickers using a pain message in inducing (a) lower purchase intentions and (b) higher intentions to reduce general meat consumption.*


Table 1 depicts all four hypotheses with their respective studies, variables and main objectives.

## 2. Method and Materials 

### 2.1. General Experimental Design

To test our proposed hypotheses, a total of four Qualtrics surveys (two pre-tests and two main studies) were conducted using online respondents via Prolific. In a break from prior studies on anthropomorphism and denial of animal mind, we worked with on-pack stickers, an inspiration from the anti-smoking stickers used on cigarette packaging. As a recent meta-analysis on such negative stickers revealed that pictures are extremely powerful and are more effective than text-only warnings [50] in decreasing intentions to start smoking and increasing intentions to quit smoking, we decided to use both text and pictures in the manipulations of our studies.

In pre-test 1 we asked respondents to rate our custom made on-pack stickers on their respective anthropomorphic traits (i.e., intelligence, pro-social behavior and pain) as well as rate these stickers on their capacity to elicit both positive and negative emotions. Following satisfactory results from pre-test 1 we launched our first main study. Study 1 exposed respondents to a random package (beef burger patties) exhibiting either an anthropomorphic or non-anthropomorphic sticker then asked them to fill specific scales related to H1, H2, and H3. Following the completion of study 1 we launched pre-test 2 which measured the same variables as pre-test 1 but on a new set of stickers. These new stickers were custom made to reflect negative anthropomorphism for each anthropomorphic trait, specifically those of intelligence and pro-social behavior which are usually depicted in a positive light. In the view of the results obtained in pre-test 2 we launched our second main study. Study 2 was similar to study in layout and variables measured but the all negative anthropomorphic traits stickers allowed us to test H4. 

### 2.2. Experiments 

#### 2.2.1. Pre-Test 1

We designed three stickers (Figure 1) each reflecting one of the anthropomorphic traits we wanted to test: intelligence, pro-social and pain anthropomorphism. All stickers are in black and white and show pictures of cow (s) with a small text under the photos. The text and the cow photo are designed to reflect the same anthropomorphic idea. The text is in white on a black background while the website is fictitious and in red font on the same black background. Fifty respondents were recruited via Prolific (66% Males, *M*_age_ = 24.38, *SD*_age_ = 5.27) and rated all the stickers, presented in random order, on their capacity to enhance the belief that cows are emotional, pro-social and intelligent creatures and to elicit positive emotions and negative emotions.

Cows’ capacities to be intelligent creatures, pro-social creatures and emotional creatures were each measured on a single item, seven-point scale (to what extent could this sticker lead to consider cows as more “intelligent creatures, “pro-social creatures” and “emotional creatures”). We measured capacity to elicit positive emotions with a two-item, seven-point scale “Please indicate to what extent the picture in the sticker ‘elicited positive emotions in you’ and ‘made you feel positive’” (1 = Not at all, 7 = Very much so). The two-items were averaged into a new variable with Cronbach’s α = 0.938. Similarly, capacity to elicit negative emotions was measured with a two-item, seven-point scale “Please indicate to what extent the picture in the sticker “elicited negative emotions in you” and “made you feel negative’” (1 = Not at all, 7 = Very much so). The two-items were averaged into a new variable with Cronbach’s α = 0.941. 

#### 2.2.2. Study 1

We recruited two hundred ninety-four respondents via Prolific, 19 participants were removed due to attention check failures (as an attention check, participants were asked to select the “Strongly Agree” option in one of the scales’ items. Those who failed to do so were removed from the analysis) leaving us with 275 submissions (64% Males, *M*_age_ = 26.21, *SD* = 9.76). After providing their consent, respondents were told that they will be seeing a burger product which they needed to evaluate. They were informed that there are no right or wrong answers but that only their opinions matter. Each respondent was randomly assigned to one of five conditions and saw a package of a product with an on-pack sticker (or no sticker, depending on the condition, see further). There was no option to go back to view the product after they decide to move to the next part of the survey. In each condition, the participant saw a package containing four beef burger patties; pending the condition the package either had no anthropomorphic sticker (control package), a sticker showing only a photo of a cow, an intelligence anthropomorphic sticker, a pro-social anthropomorphic sticker, or a pain anthropomorphic sticker (see Figure 2). We also included a sticker with only a photo of a cow stressing the meat-cow connection to rule out that it is the mere connection of linking meat to a cow that can explain our results [51,52] rather than stressing anthropomorphic characteristics of the cow. After exposure to the beef burger patties, respondents proceeded to fill out several scales (purchase intention, intention to reduce meat consumption, attitudes towards meat and anticipatory guilt).

We measured purchase intention with a single item seven-point scale “How likely would you be to purchase the ‘burger beef patties’ that you just saw?” (1 = Not at all likely, 7 = Very likely) based on the work of McCall and Lynn [53]. Intention to reduce meat consumption due to packaging was measured via a three-item, seven-point scale “This packaging could motivate me to reduce my beef consumption”, “This packaging could help me eat less beef” and “After seeing this packaging, I would like to reduce my beef consumption” (1 = Strongly disagree, 7 = Strongly agree, Cronbach’s α = 0.94) adapted from the work of Adams et al. [54]. Attitude towards meat was measured with a modified two item, seven-point scale adapted from Raghunathan et al. [55] “How tasty do you think the “burger beef patties” would be?” and “How much do you think you would enjoy eating the “burger beef patties”?” (1 = Not at all, 7 = Very much, Cronbach’s α = 0.86). Anticipatory guilt was measured with a four item, seven-point scale adapted from Cotte et al. [56] “Imagine you are now eating the ‘burger beef patties’, please indicate how would you feel: ‘guilty’, ‘responsible’, ‘accountable’ and ‘ashamed’ (1 = Not at all, 7 = Very much, Cronbach’s α = 0.84). 

#### 2.2.3. Pre-Test 2

Fifty respondents recruited via the Prolific website took part in our pre-test (72% Males, *M*_age_ = 23.22, *SD* = 4.63). Each respondent was randomly exposed to each of the three anthropomorphic conditions (negative pro-social sticker, negative intelligence sticker and the pain sticker). The stickers maintained their format from our prior experiments (black and white cow photo + text). We used the same photo for all three conditions while the text is reflective of the new negatively oriented pro-social and intelligence anthropomorphisms. Figure 3 shows all three negative anthropomorphic stickers.

The extent to which cows are emotional creatures was measured by a one item seven-point scale (not at all-very much so), the same was done for the pro-social creatures’ question and intelligence creatures’ question. The capacity of the picture in the sticker to elicit positive emotions was measured with a two-item, seven-point scale (Cronbach’s α = 0.886) (To what extent the picture in the sticker: “elicited positive emotions in you?” and “made you feel positive”?), similarly the capacity of the picture in the sticker to elicit negative emotions was also measured with a two-item, seven-point scale (Cronbach’s α = 0.872) (To what extent the picture in the sticker: “elicited negative emotions in you?” and “made you feel negative”?).

#### 2.2.4. Study 2

Two hundred eighty-one respondents recruited via the Prolific website took part in our survey, six participants were removed due to attention check failures leaving us with 275 submissions (63% Males, *M*_age_ = 25.22, *SD* = 7.81). Each respondent was randomly assigned to one of the five conditions similar in Study 1, except now we are using the new negatively aligned anthropomorphic stickers. After exposure to the beef burger patties, respondents proceeded to fill out the same scales as in the first study.

## 3. Results and Discussion 

### 3.1. Pre-Test 1

We first checked whether our manipulation was successful. That is, we checked whether respondents who were exposed to a specific sticker indeed rate cows higher on the respective anthropomorphic trait than respondents exposed to another sticker. A repeated measures ANOVA taking “sticker” as within factor and their respective anthropomorphic traits as dependent variables showed that the intelligence sticker is rated significantly higher on cows being intelligent creatures than the pain sticker (*M*_intelligence_ = 4.96, *SD* = 1.65, *M*_pain_ = 2.82, *SD* = 1.51, *p* < 0.05) and the pro-social sticker (*M*_pro-social_ = 3.16, *SD* = 1.67, *p* < 0.05. Similarly, the pro-social sticker scores significantly higher on cows being pro-social creatures than the pain sticker (*M*_pro-social_ = 5.56, *SD* = 1.51, *M*_pain_ = 3.76, *SD* = 1.78, *p* < 0.05) and the intelligence sticker (*M*_intelligence_ = 3.64, *SD* = 1.79, *p* < 0.05). Also, the pain sticker scores significantly higher on cows being emotional creatures than the pro-social (*M*_pain_ = 5.86, *SD* = 1.40, *M*_pro-social_ = 4.6, *SD* = 1.96, *p* < 0.05) and intelligence stickers (*M*_intelligence_= 4.16, *SD* = 1.81, *p* < 0.05).). Our manipulation thus proves successful. Next, we checked the emotions the different stickers evoked. Again, repeated measures ANOVA were conducted. Results revealed that the intelligence and pro-social sticker score significantly higher on positive emotions than the pain sticker (*M*_pro-social_ = 3.68, *SD* = 1.63, *M*_intelligence_ = 3.37, *SD* = 1.55, *M*_pain_ = 2.35, *SD* = 1.24, *p’s* < 0.05) while the latter scores significantly higher on negative emotions (*M*_pro-social_ = 3.25, *SD* = 1.55, *M*_intelligence_ = 3.31, *SD* = 1.61, *M*_pain_ = 4.48, *SD* = 1.70, *p’s* < 0.05). This is in line with our assessment of the valence of anthropomorphic traits (some are inherently more positive than others) and how respondents perceive them in animals. Giving the results of the pre-test we decide to use these stickers in our main study 1.

### 3.2. Study 1 

A one-way analysis of variance (ANOVA) taking package (control, just cow, and the three anthropomorphic stickers) as independent variable and purchase intention as dependent variable showed a significant effect (*F* (4270) = 6.317, *p* < 0.05, *η*^2^ = 0.086). Similar results were also observed for intentions to reduce meat consumption *F* (4270) = 8.575, *p* < 0.05, *η*^2^ = 0.113. 

A first planned contrast test revealed that the anthropomorphic conditions (i.e., intelligence, pro-social, pain) as compared to the non-anthropomorphic conditions (i.e., control, just cow) induced significantly lower purchase intentions (*t* (256.983) = −3.607, *p* < 0.05), while they significantly enhanced the belief that the package could reduce meat consumption (*t* (270) = 5.085, *p* < 0.05). These results provide support for H1. 

To test H2, we ran another planned contrast in which we compared the positive trait conditions (pro-social and intelligence) with the negative trait condition (pain). The results showed that the negative trait (vs. positive traits) resulted in significantly lower purchase intentions (*t* (90.199) = −2.766, *p* = 0.007) and a higher belief that the package is able to reduce meat consumption (*t* (270) = 2.487, *p* = 0.014). These results are displayed in Figure 4 and Figure 5.

A breakdown of planned contrasts comparing each positive (i.e., intelligence and pro-social) anthropomorphic sticker to pain sticker revealed a marginal significant difference between pain and pro-social anthropomorphism on purchase intentions (*t* (104.866) = −1.815, *p* = 0.072) and intention to reduce meat consumption (*t* (270) = 1.851, *p* = 0.065) while the intelligence sticker was significantly different than pain sticker on purchase intention (t (98.472) = −3.159, *p* = 0.020) and intentions to reduce meat consumption (*t* (270) = 2.469, *p* = 0.014). 

To further test H2, we also compared the individual anthropomorphic conditions with the non-anthropomorphic animal condition (i.e., just cow). Only pain showed a significant advantage over and above reminding participants of the animal origin and this for both dependent variables (purchase intentions: *t* (98.070) = −3.104, *p* = 0.02; intentions to reduce meat consumption due to packaging: *t* (270) = 3.947, *p* < 0.05). Hence, we conclude that our results provide support for H2. 

For H3, we conducted a mediation analysis using the PROCESS model 6 macro for SPSS [57] to test whether respondents anticipatory guilt and attitudes towards meat mediated the effect of anthropomorphism on their purchase intentions. We dummy coded the conditions as follows: 0 = control and just cow conditions, 1 = all 3 anthropomorphic traits, we entered anticipatory guilt as first mediator and attitude towards meat as second mediator. A bias-corrected bootstrap analysis with 5000 samples and 95% bias-corrected intervals (CIs), indicated a significant indirect effect of anthropomorphism on purchase intentions via anticipatory guilt and attitudes towards meat (*β* = −0.135, *SE* = 0.05, 95% *CI* [−0.250, −0.043], we also observed a significant direct effect of anthropomorphism vs. no anthropomorphism on purchase intentions (*β* = −0.365, *SE* = 0.16, 95% *CI* [−0.686, −0.044]. We can conclude that the impact of anthropomorphism on purchase intentions of meat was partially mediated by anticipatory guilt and attitudes towards meat. As such, the data provide support for H3. A similar mediation analysis on individual anthropomorphic traits vs. no anthropomorphism revealed a significant indirect effect of pain anthropomorphism on purchase intentions via anticipatory guilt and attitudes towards meat (*β* = −0.166, *SE* = 0.08, 95% *CI* [−0.343, −0.036]). However, no such serial mediation was observed for intelligence anthropomorphism (*β* = −0.093, *SE* = 0.07, 95% *CI* [−0.254, 0.072]) or pro-social anthropomorphism (*β* = −0.840, *SE* = 0.06, 95% *CI* [−0.223, 0.016]) because both traits lacked a significant *a*_1_ path (conditions’ impact on anticipatory guilt) in their serial mediation (see Figure 6). This is also corroborated by planned contrast tests comparing individual anthropomorphic traits (vs. non-anthropomorphic conditions) on anticipatory guilt which revealed a significant difference between the pain trait condition and the non-anthropomorphic conditions (*t* (270) = 3.788, *p* = 0.002), while the difference between the intelligence trait and the non-anthropomorphic conditions was only marginally significant (*t* (270) = 1.748, *p* = 0.082) and the difference between the pro-social trait condition and the non-anthropomorphic conditions was non-significant (*t* (270) = 1.572, *p* = 0.117).

As predicted by H1, consumers exposed to anthropomorphic stickers vs. non-anthropomorphic conditions had reduced purchase intentions and a greater intention to reduce meat consumption. This result is in alignment with Chan’s [37] proposed usage of anthropomorphism as a conservation tool for animals. 

The observed impact on purchase intentions by our operationalization of pro-social anthropomorphism (capacity to form families) goes beyond the results of Wang and Basso’s [21]. These researchers only observed an impact of pro-social anthropomorphism, which they operationalized as friendship between animals, when pigs were the focal animal of their manipulation but not when cows were used. It remains unclear why our results were different, but it should be noted that not only our operationalization of pro-social anthropomorphism was different but also our manipulation (on-pack stickers) was different than that of Wang and Basso’s [38] manipulation (a meat vendor’s webpage describing their cows). 

When comparing pain anthropomorphism to the positive intelligence and pro-social anthropomorphism we observed a lower intention to purchase the meat product and a bigger intention to reduce meat consumption; these findings confirmed H2. A one-on-one comparison of the anthropomorphic traits with pain anthropomorphism revealed a marginal difference between pro-social and pain on purchase intentions and intentions to reduce meat consumption while it was significant between pain and intelligence. However, when compared against the non-anthropomorphic cow condition only pain showed significant advantage over reminding participants of the meat origins. These results highlight the effectiveness of using pain anthropomorphism in comparison to pro-social behavior and intelligence anthropomorphism as well as non-anthropomorphic reminder of meat origins (i.e., just cow picture) in affecting purchase intentions and intentions to reduce meat consumption. Kunst and Hohle [52] argued that denial of animal mind in consumers works in tandem with dissociation (i.e., meat is different than the animal that provided the meat) but that denial of animal mind only arises when consumers are told explicitly that their behavior contributes to the death or suffering of animals similar to what our experimental manipulation insinuated with the anthropomorphic stickers. 

Anthropomorphic messages’ impact on purchase intentions was shown to be serially mediated by anticipatory guilt and attitude towards meat, thus providing support for H3. Individual anthropomorphic traits mediation analysis revealed that only pain (vs. non-anthropomorphic conditions) anthropomorphism’s impact on purchase intentions is serially mediated by anticipatory guilt and attitude towards meat. However, no such serial mediation was found for pro-social or intelligence anthropomorphism. These results seem to suggest that pain anthropomorphism was able to elicit higher levels of guilt than the positive anthropomorphic traits. On a final note, it must be pointed out that Wang and Basso [38] did not attempt a mediation analysis with their cows’ study seeing there was no main effect of cows’ friendship anthropomorphism on purchase intentions and that their observed serial mediation was conducted solely on the pig condition. 

### 3.3. Pre-Test 2

A repeated measure ANOVA with “Positivity” as the within-subject factor was conducted. Estimated marginal means using Bonferroni correction revealed no significant difference between all three anthropomorphic stickers (*M*_pro-social_ = 2.2, *SD* = 1.46, *M*_intelligence_ = 2.33, *SD* = 1.42, *M*_pain_ = 2.15, *SD* = 1.44, *p’s* > 0.05). Similarly, no difference was detected between all three stickers on their negativity rating when “Negativity” is taken as the within-subject factor (*M*_pro-social_ = 4.76, *SD* = 1.71, *M*_intelligence_ = 4.52, *SD* = 1.65 and *M*_pain_ = 4.76, *SD* = 1.74, *p’s* > 0.05). We observe that when intelligence and pro-social anthropomorphism are painted in a negative light their negativity is now similar to that of pain anthropomorphism. Likewise, all three traits score low on the positivity rating. Similar to study 1’s pretest, we checked if the stickers do reflect the anthropomorphic traits they are designed to elicit. As such a repeated measure ANOVA with “emotional creatures rating” as the within-subject factor was conducted. Estimated marginal means using Bonferroni correction revealed that the pain anthropomorphism is significantly different than the intelligent anthropomorphism (*M*_pain_ = 5.56, *SD* = 1.57, *M*_intelligence_ = 5.1, *SD* = 1.49, *p* < 0.05) but not than the pro-social anthropomorphism (*M*_pro-social_ = 5.86, *SD* = 1.12, *p* = 0.34). Similarly, the test was repeated for the “pro-social creatures rating” and the “intelligent creatures rating” revealing that pro-social anthropomorphism is significantly different than pain and intelligent anthropomorphism on pro-social creatures rating (*M*_pro-social_ = 5.54, SD = 1.51, *M*_pain_ = 3.94, *SD* = 1.92 and *M*_intelligence_ = 4.72, SD = 1.73, *p’s* < 0.05) and that intelligent anthropomorphism is significantly different than pain and pro-social anthropomorphism on intelligent creatures rating (*M*_intelligence_ = 5.14, *SD* = 1.59, *M*_pain_ = 4.28, *SD* = 1.82, and *M*_pro-social_ = 4.54, *SD* = 1.7, *p’s* < 0.05). According to Gilam et al. [58], pain is defined as a disagreeable personal experience with a sensory and an emotional element. It can be argued that both pain and pro-social stickers are indeed reflecting their respective anthropomorphic values but the grief and loss experienced by separation from one’s young causes (as displayed in the pro-social anthropomorphism sticker) more discomfort than the thought of physically being hurt (as displayed in the pain anthropomorphism sticker). This said, a one sample *t*-test reveals that both stickers score significantly above the “emotional creatures scale’s” midpoint of 4 (*M*_pain_ = 5.56, *SD* = 1.57, *t* (49) = 7.039, *p* < 0.05 and *M*_pro-social_ = 5.86, *SD* = 1.12, *t* (49) = 11.69, *p* < 0.05). We decide to proceed with the stickers for our study 2. 

### 3.4. Study 2 

We conducted a one-way analysis of variance (ANOVA) taking package (control, just cow, and the three anthropomorphic stickers) as independent variable and purchase intentions as dependent variable showed a significant effect (*F* (4,270) = 3.288, *p* = 0.012, *η*^2^ = 0.046). A similar result was also observed for intentions to reduce meat consumption *F* (4,270) = 9.978, *p* < 0.05, *η*^2^ = 0.129.

Similar to study 1, a planned contrast test revealed that the anthropomorphic conditions (i.e., intelligence, pro-social and pain) as compared to the non-anthropomorphic conditions (i.e., control, just cow) induced significantly lower purchase intentions (*t* (247.386) = −3.281, *p* = 0.001) while they significantly enhanced the belief that the package could reduce meat consumption (*t* (270) = 5.725, *p* < 0.05) These results replicate the ones of study 1 and provide further support for H1. 

To test H4, we ran another planned contrast in which we compared the pain anthropomorphic condition with the negative pro-social and intelligence anthropomorphic conditions. The results showed no difference between the three anthropomorphic traits on neither of the two dependent variables, *p*’s > 0.05. A breakdown of planned contrasts comparing each anthropomorphic sticker to each other revealed no difference between the three anthropomorphic traits for neither of the dependent variables, *p*’s > 0.05. These results provide support for H4 (see Figure 7 and Figure 8). 

To further test H4, we also compared the individual anthropomorphic conditions with the non-anthropomorphic animal condition. None of the three anthropomorphic conditions differed from the just-cow condition on purchase intentions, *p*’s > 0.05. However, pain (*t* (270) = 2.857, *p* = 0.005) and pro-social (*t* (270) = 2.901, *p* = 0.004) conditions were significantly better than just-cow condition in their capacity to increase intentions to reduce meat consumption while intelligence condition was marginally so (*t* (270) = 1.760, *p* = 0.080). 

We again tested H3 by conducting a mediation analysis using the PROCESS model 6 macro for SPSS to check whether respondents’ anticipatory guilt and attitudes towards meat mediated the effect of anthropomorphism on their purchase intentions. We dummy coded the conditions as follows: 0 = control and just cow conditions, 1 = all three anthropomorphic traits, we entered anticipatory guilt as first mediator and attitude towards meat as second mediator. A bias-corrected bootstrap analysis with 5000 samples and 95% bias-corrected intervals (CIs), indicated a significant indirect effect of anthropomorphism on purchase intentions via anticipatory guilt and attitudes towards meat (*β* = −0.1776, *SE* = 0.05, 95% *CI* [−0.2897, −0.0787], we also observed a significant direct effect of anthropomorphism vs. no anthropomorphism on purchase intentions (*β* = −0.4461, *SE* = 0.17, 95% *CI* [−0.7817, −0.1106] (see Figure 9). 

We conclude that the impact of anthropomorphism on purchase intentions of meat was partially mediated by anticipatory guilt and attitudes towards meat, providing support for H3. A similar mediation analysis on pain anthropomorphic trait vs. no anthropomorphism revealed a significant indirect effect of pain anthropomorphism on purchase intentions via anticipatory guilt and attitudes towards meat (*β* = −0.2599, *SE* = 0.10, 95% *CI* [−0.4780, −0.0772]). Similar results were observed for pro-social negative anthropomorphism (*β* = −0.1920, *SE* = 0.09, 95% *CI* [−0.3797, −0.238]) and negative intelligence anthropomorphism (*β* = −0.2132, *SE* = 0.10, 95% *CI* [−0.4057, −0.0604]). Unlike study 1, where the *a*_1_ path (conditions’ impact on anticipatory guilt) was non-significant for both intelligence and pro-social anthropomorphism, we observed here that all three conditions instigated a high level of guilt. A planned contrast of each individual anthropomorphic traits (vs. non-anthropomorphic conditions) revealed a significant difference on anticipatory guilt (Pain: *t* (270) = 3.032, *p* = 0.003; Pro-social: *t* (270) = 2.122, *p* = 0.035; Intelligence: *t* (270) =2.717, *p* = 0.007). 

In study 2, we were able to replicate the results of study 1 by showing that anthropomorphic traits (vs. non-anthropomorphic conditions) on packaging do indeed affect purchase intentions and intentions to reduce meat consumption in the hypothesized direction thus again providing support for H1. 

We also tested H4 where pro-social and intelligence anthropomorphism were formulated in a negative way. We argued that capacity to experience pain is inherently perceived as negative while pro-social and intelligence capacities are seen as positive, which was observed in the pretest of study 1. Negatively formulated pro-social and intelligence capacities operationalized as “trauma when losing loved ones” and “denial of mental stimulation” respectively were shown to be similar in valence to a pain message depicting general capacity to suffer. All three anthropomorphic traits have similar effects on purchase intentions and intentions to reduce meat consumption. These findings support H4 and shed an interesting light on how information in an anthropomorphic context can have different effects depending on its positive or negative formulation. Comparing each anthropomorphic trait to just-cow condition did reveal unexpected results. Purchase intention was similar between the three anthropomorphic traits and the non-anthropomorphic just cow condition, but intentions to reduce to reduce meat consumption due to packaging was significantly higher for pain and pro-social anthropomorphism but only marginally so for intelligence anthropomorphism. Similar to study 1, we observed that anthropomorphism effects over just-cow (i.e., reminding people of the meat’s origin) is not superior in all measured variables but still significant on several measures. 

Finally, we retested H3 and found similar results to study 1 where the impact of anthropomorphic messages (vs. non-anthropomorphic conditions) on purchase intentions was serially mediated by anticipatory guilt and attitude towards meat. However, unlike study 1 where only pain as an individual anthropomorphic message (vs. non-anthropomorphic conditions) exhibited such mediation, we observe here the same results for the negatively formulated pro-social and intelligence anthropomorphism. The negative formulation enabled pro-social and intelligence stickers to elicit more anticipatory guilt than their positively formulated counterparts. These results give further support to the impact of the negativity bias on anthropomorphism and by extension purchase intentions. 

## 4. General Discussion

### 4.1. Summary of Findings

The current research establishes that the usage of anthropomorphism via on-pack stickers can lead to decreased purchase intentions of beef-burger patties along with an intention to decrease meat consumption in general. Several planned contrasts in study 1 revealed that the individual anthropomorphic stickers tend to outperform the control package (no sticker) but only pain anthropomorphism was able to outperform the non-anthropomorphic cow condition (just-cow sticker) in its capacity to reduce purchase intentions of the burger beef-patties and increase intentions to reduce general meat consumption. Mediation analysis showed that anthropomorphism’s (vs. non-anthropomorphic conditions) impact on purchase intentions was serially mediated by anticipatory guilt and attitude toward meat. A similar mediation result was observed for the pain anthropomorphism due to its ability to solicit more anticipatory guilt than pro-social and intelligence stickers. In our second experiment, we again observed how on-pack stickers using anthropomorphized animals lead to decreased purchase intentions of our product and a general intention to decrease meat consumption. We also demonstrated how formulating intelligence and pro-social anthropomorphic conditions in a negative light allowed them to have similar results to pain anthropomorphism. A serial mediation for each individual anthropomorphic trait (vs. non-anthropomorphic conditions) is observed due to increased levels of anticipatory guilt; this is different from the results of study 1 where neither pro-social or intelligence anthropomorphisms exhibited the serial mediation. Lastly, we observed a divergence between the pain anthropomorphism of study 1 and study 2. A pain sticker was able to significantly reduce purchase intentions more than the just-cow condition in study 1 but was unable to do so in study 2. Furthermore, the same results were observed for the negatively formulated pro-social and intelligence anthropomorphism. 

### 4.2. Theoretical and Practical Contributions 

Meat is a staple food item in many diets all over the world. Meat’s established dietary status in the eyes of consumers afford it much motivated cognition to justify its continued demand and consumption regardless of the cost it incurs on public health, the environment or the welfare of animals [2,29,59]. Prior studies on barriers to reducing meat consumption or adopting a meatless diet revealed that health concerns and meat enjoyment are primary reasons why consumers find it hard to reduce or quit the meat habit [60,61,62]. Regardless of these barriers, continued meat consumption can cause its consumers to experience cognitive dissonance making the reduction of this state a priority for any meat-eater. Barriers to MRCD and strategies designed to reduce MRCD [16] employed by meat-eaters can help policy makers develop new approaches to combat this motivated cognition. In the context of a modern supermarket, where most food shopping happens in developed countries, the origin of the meat is often erased by dissociation. Meat becomes a product divorced from its animal origin. Our approach to use on-pack stickers not only helps counteract the act of dissociation but also provides a solution to a well-established strategy of meat-eating justification i.e., “denial of animal mind” by using a sticker with an anthropomorphic message. Anthropomorphism’s ability to assign human like characteristics to non-human agent offers a strong natural counter argument to the denial of animal mind. Moreover, a negative anthropomorphic message like the capacity to feel pain and suffering taps into the phenomena of negativity bias. From our experiments, we argued that negative anthropomorphism operationalized with a sad looking cow and a text highlighting cows’ capacity to experience pain and suffering tapped into the negativity bias and triggered more anticipatory guilt than pro-social and intelligence anthropomorphisms. The impact of negativity bias can also be extended to pro-social and intelligence anthropomorphic messages by reformulating them in a negative light. The new negative pro-social and intelligence anthropomorphisms outperformed their positive counterparts in impacting meat consumption and mimicked pain anthropomorphism in eliciting anticipatory guilt. These findings offer a new path in which anthropomorphism and meat consumption can be pursued in future research where anthropomorphic traits are similar in their impact on meat consumption if negativity bias is used as the norm in describing animals’ similarity to humans.

### 4.3. Limitation and Future Directions

In this paper, we tested only one product (beef patties) along with one animal, i.e., cow. It would be interesting to see if such results can extend to other animals like pigs and chickens, the most consumed type of meats worldwide. Chickens would offer an interesting extension on this line of research seeing that unlike cows they are less phylogenetically related to humans and as such attributed less empathy and anthropomorphic tendencies [63]. 

From the consumer side, meat-eaters have been shown to come in different meat eating commitment levels [64], animal and environmental welfare conscientiousness [65], environmental values [66], and different cultural views on animals [67]. Our intervention was conducted on a random prolific sample where the type of meat-eater was never controlled for, a future research could explore the effectiveness of negativity bias on different types of meat-eaters. On a similar note, this research was conducted online and lacked an actual behavioral choice, while purchase intentions and intentions to reduce meat consumption were successfully manipulated by the usage of pain anthropomorphism, it remains to be seen if this effect would carry through to a real-life shopping environment. 

Another limitation in our current work is the explicit nature of our measured variables. These explicit measures are usually susceptible to impression management (i.e., social desirability) or lack introspective accuracy (i.e., thoughts and feelings that might be outside the conscious awareness) which limits their practical benefits to researchers [68]. A possible solution would be the usage of implicit measures which are less susceptible to such issues [69]. Past research on meat using implicit measures focused on the known-group approach where attitudes towards meat and vegetables were compared between two groups (meat eaters vs. vegetarians) using different implicit measurement tools like the implicit association task [70] and the implicit relational assessment procedure [71]. A more recent work by Love and Sulikowski [72] compared men and women on their implicit attitudes towards healthiness of meat operationalized in terms of virility and strength. But to our knowledge, no implicit measurement study was conducted where animal anthropomorphism and meat consumption were the focal point of the research. Next to implicit measures also neuroscience tools such as eye tracking, electroencephalography (EEG) and galvanic skin response (GSR) [73,74] could be used to investigate the ability of different types of stimuli to evoke a neurological response related to decreasing meat eating intentions. Rilling et al. [75] using functional magnetic resonance imaging (fMRI) investigated cooperation based reciprocal altruism in a sample of 36 women which was found to be linked to reward centers in the human brain. It could be interesting to investigate whether anthropomorphic (vs. non-anthropomorphic) stimuli or pain-based (vs. socially or intelligence based) anthropomorphism are differentially linked to activation of reward areas which could reinforce reciprocal altruism, thereby motivating people to have lower meat-eating intentions We encourage colleagues to test the value of different anthropomorphic stimuli using both implicit measures and neuroscience tools.

Anthropomorphism is introduced as a natural remedy for denial of animal mind [16]. However, in our current work we did not measure the degree to which mind was attributed to cows. The difference in mind attribution could possibly explain the difference in study 1 and 2 between anthropomorphic traits and the non-anthropomorphic cow condition on purchase intentions. When does anthropomorphism perform better than just reminding consumers of the animal origin? A final possible question that we would like to raise is the persistency of the effect of negative anthropomorphic stickers on meat products, would they carry through for long periods or consumers will learn to ignore them and stop being influenced by their message? 

## 5. Conclusions

Anthropomorphism has shown promise in changing consumers’ habits [34,36,37]. We added insights on the impact of using anthropomorphism in the domain of meat-eating reduction. Previous research showed that intelligence of animals perceived as food was ignored by consumers [38] and stressing pro-social behavior was seemingly only effective at curbing consumption in one animal (i.e., pigs) while being ineffective in eliciting the same impact in another (i.e., cows) [21]. Our work further explored the use of these anthropomorphic traits using on-pack stickers (using beef burger patties as product) and adds to the literature by testing the effects of an untapped anthropomorphic trait: the capacity to feel pain. This capacity is inherently associated with negative emotions which taps into the negativity bias, a natural tendency in humans to assign greater weight to negative events than they do to positive ones [22,23]. The current work showed that indeed the usage of anthropomorphic messages is effective at reducing meat eating intentions. Further testing revealed that a negatively formulated anthropomorphic message (capacity to feel pain) was capable of curbing meat-eating intentions and was more effective in doing so than the established anthropomorphic traits (intelligence and pro-social behavior) that are usually depicted in a positive light. This capacity to reduce meat eating intentions was also present when other anthropomorphic traits were painted in a negative light (the aforementioned pro-social behavior and intelligence). Furthermore, we show that anthropomorphism was able to reduce meat purchase intentions by eliciting feelings of anticipatory guilt that negatively impacted attitude towards meat. 

## Figures and Tables

**Figure 1 foods-10-02147-f001:**
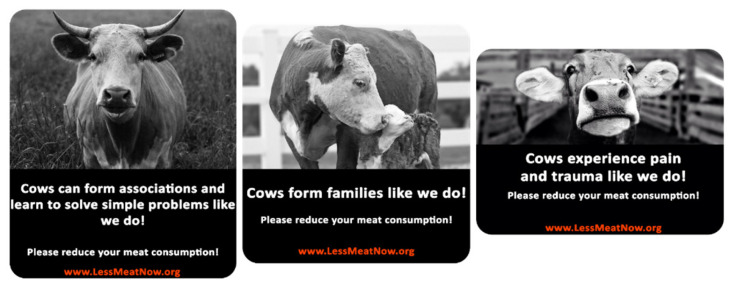
The three anthropomorphic stickers (Sticker 1: Intelligence anthropomorphism, sticker 2: Pro-social anthropomorphism, sticker 3: Pain anthropomorphism).

**Figure 2 foods-10-02147-f002:**
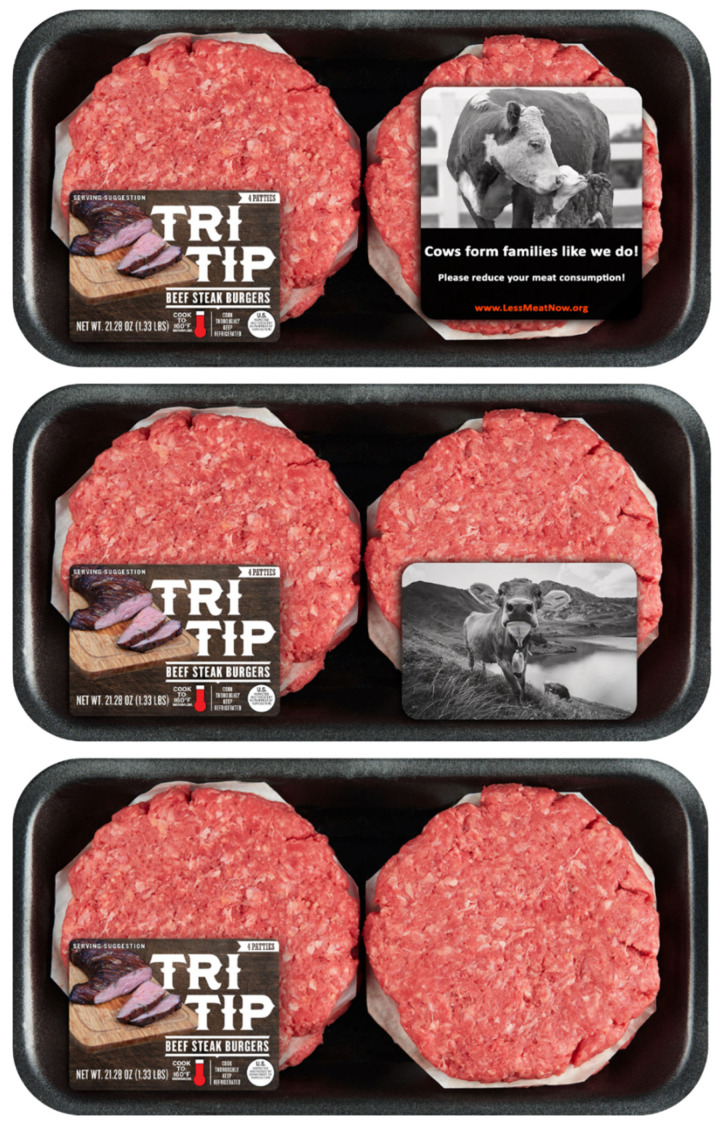
The burger beef patties product (**Bottom**: Original packaging, **Middle**: Just cow packaging, **Top**: Example of anthropomorphic sticker packaging—Pro-social anthropomorphism).

**Figure 3 foods-10-02147-f003:**
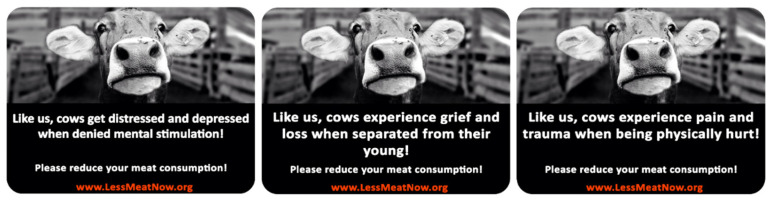
The three anthropomorphic stickers (Sticker 1: Negative intelligence anthropomorphism, sticker 2: Negative pro-social anthropomorphism, sticker 3: Pain anthropomorphism).

**Figure 4 foods-10-02147-f004:**
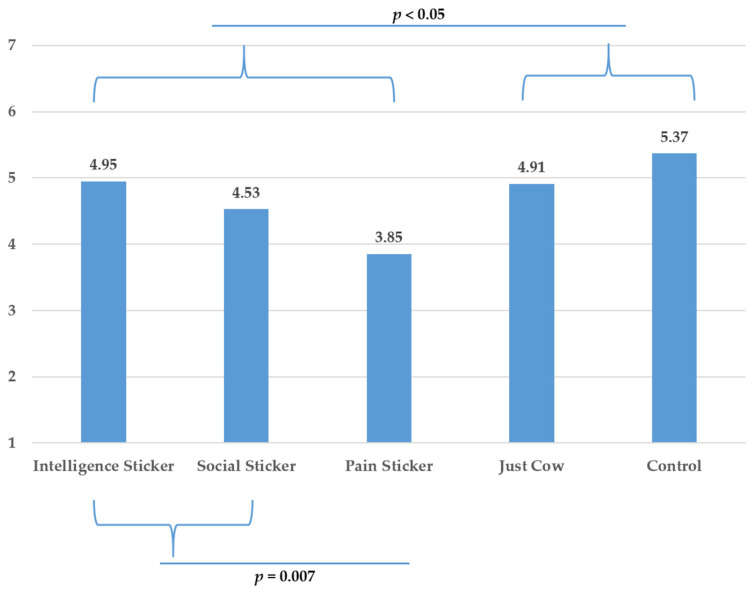
Purchase Intentions: Means for the different conditions and results contrast analyses (Study 1).

**Figure 5 foods-10-02147-f005:**
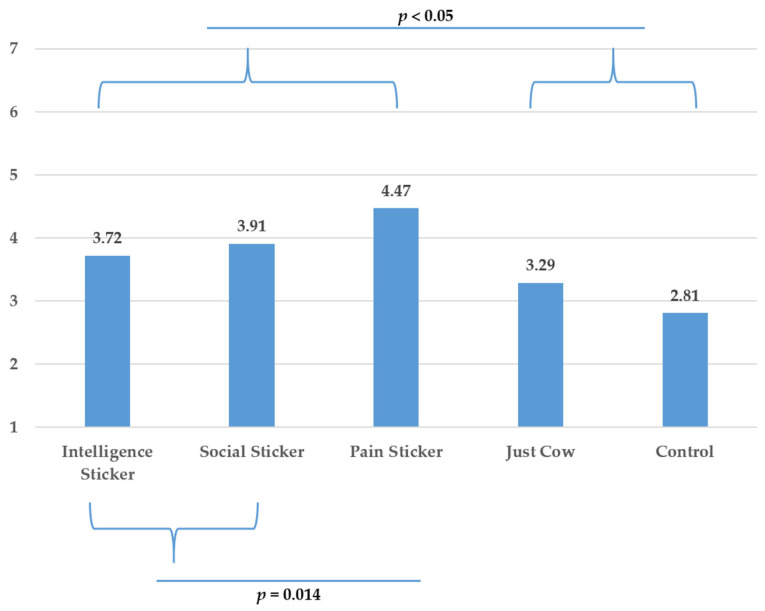
Intentions to reduce meat consumption due to packaging: Means for the different conditions and results contrast analyses (Study 1).

**Figure 6 foods-10-02147-f006:**
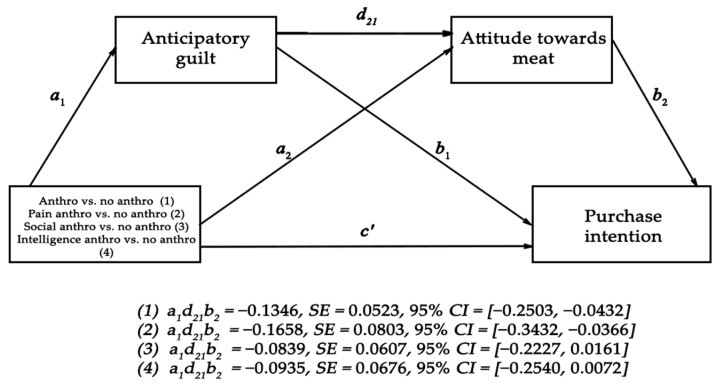
The effect of anthropomorphism (intelligence, pro-social and pain) vs. no anthropomorphism (control and just cow) on purchase intentions is serially mediated via anticipatory guilt and attitude towards meat (Study 1).

**Figure 7 foods-10-02147-f007:**
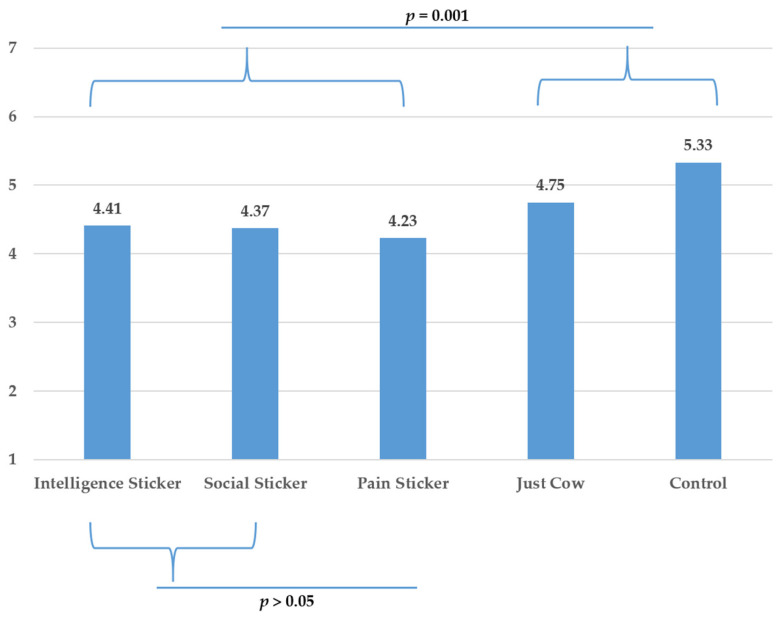
Purchase Intentions: Means for the different conditions and results contrast analyses (Study 2).

**Figure 8 foods-10-02147-f008:**
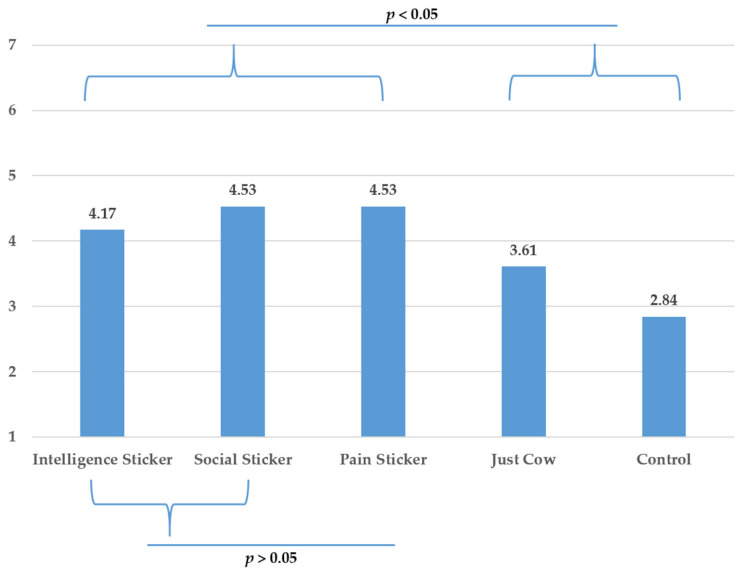
Intention to reduce meat consumption due to packaging: Means for the different conditions and results contrast analyses (Study 2).

**Figure 9 foods-10-02147-f009:**
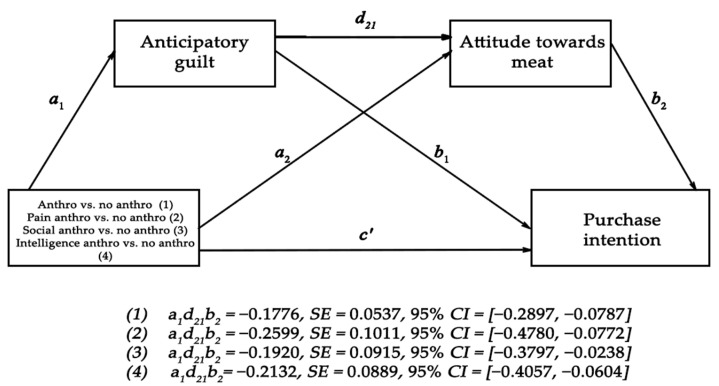
The effect of anthropomorphism (pain, pro-social and intelligence) vs. no anthropomorphism (control and just cow) on purchase intentions is serially mediated via anticipatory guilt and attitude towards meat (Study 2).

**Table 1 foods-10-02147-t001:** All studies with their respective stimuli, variables and objectives.

Study	Stimuli	Variables	Main Objective(s)
Pre-test 1	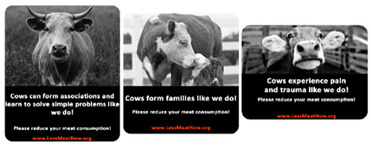	Capacity to elicit specific anthropomorphic traits Capacity to elicit positive emotions Capacity to elicit negative emotions	To check for the suitability of the stickers for study 1
Study 1	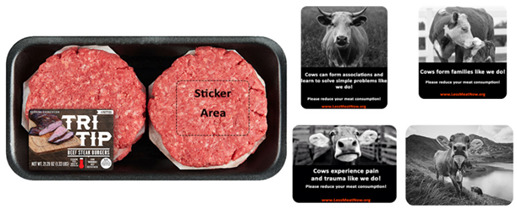 N.B: The control package has no sticker beside the original product sticker	Purchase Intention Intention to reduce meat consumption due to packaging Attitude towards meats Anticipatory guilt	To test H1, H2 and H3
Pre-test 2	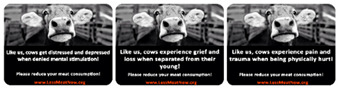	Similar to pre-test 1	To check for the suitability of the stickers for study 2
Study 2	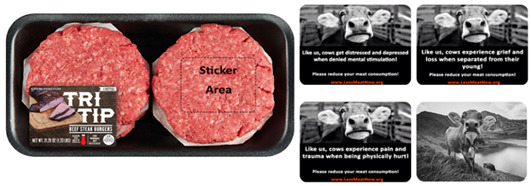 N.B: The control package has no sticker beside the original product sticker	Similar to study 1	To test H4

## Data Availability

Z.C., M.G., & I.V. (8 September 2021). Animals Like Us: Leveraging the Negativity Bias in Anthropomorphism to Reduce Beef Consumption. Retrieved from osf.io/evzjn (accessed on 8 September 2021).

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
