# Peer review of "Animals Like Us: Leveraging the Negativity Bias in Anthropomorphism to Reduce Beef Consumption"

_foods, 2021, doi:10.3390/foods10092147_

Round 1

Reviewer 1 Report

The topic analyse in this paper is very interesting and relevant, since there is an increase of interest in the impact of meat production and consumption. Authors use recent research related to the topic and establishes the originality of the research and need for investigations in the area.

Introduction

It can be improved both in clarity and development. You need to revise the introduction: is vital. I emphasize the need to rewrite the introduction to make it more engaging and to be better recognized by readers. I emphasize the need to rewrite the introduction to make it more appealing and to be better recognized by readers.  

Hypotheses should be better grounded. Hypotheses analysis could be improved. The authors may reference some theories such as the theory of planned behavior (TPB) to help explain and predict consumer behavior in purchase intention. Besides, the first statement should be written in present tense, never in future, as you did with the other three hypotheses

Method

I am not sure your stickers are strictly speaking metaphors (“ a figure of speech that describes an object or action in a way that isn't literally true, but helps explain an idea or make a comparison”). Animals do have the capacity to experience pain, and have intelligence and pro-social behavior. The  figure of style used by the authors attributing human properties to an animal, to which it is makes the cows speak or react like a person, could be better called personification or prosopopeia.

Have the scales applied in the research (purchase intention, intention to reduce meat consumption, attitude towards meat, anticipatory guilt) been applied in other studies? If so, include references.

Regarding the experiment design there is a lack of information about the design, the procedure in general.

Why the anthropomorphic stickers in Figure 1 have different size?

General discussion

In the discussion section, only one paper is mentioned (Wang and Basso [37]). It is necessary to broaden the discussion of the results in light of other previous studies

My major concern is regarding the effects of “metaphors” or personifications have been based on declarative statements. In future lines of research, it would be interesting to mention the opportunity to use implicit measures such as neurosciences tools that could provide additional insights crucial to understand consumers’ behaviour or intention. People are sometimes unable or unwilling to reveal their attitudes and opinions; an alternative way to measure attitude towards meat or purchase intention could be based on implicit measures as these collect unconscious subjects’ attitudes, analyse the automatic and reduce bias like social desirability

Conclusions

It can be expanded to ensure readers are presented with all the general information about the topic.

There are some typos and some sentences are still not easy to read. maybe a final edit would help.

 Please proofread all the text and make sure to correct any grammar and spelling mistakes.

  • “Our pretest revealed that s intelligence”
  • similar effectson purchase
  • on purcahse intention
  • anthropomporhism

Somo acronynms are not previously introduced (DV, PI, IR)

Reviewer 2 Report

The authors presented interesting data about the application of anthropomorphism to reduce beef consumption. In this study, the sample size (> 200 per study) is good, the statistical methods employed are mostly sound, and the results are well explained by robust referencing.  However, some improvements should be made in the structure of the paper to ease readability and clarify the logical sequence of research questions. Detailed comments are as follows:

  1. First of all, I’d like to raise a question about the practicality of anthropomorphism application with on-pack stickers by beef producers. I’m not sure whether the question/limitation presented in lines 628-631 is related to my question or not. If not, the forthcoming barrier to real-life application of the findings in the present study should be justified or explained as a limitation.
  2. Introduction: Under 1.5 section, research aims should be revised. In this research, two studies are presented. Each of the two studies has a pretest and a main test/study and in each test, H1, H2, and H3 hypotheses are tested. If so, research aims should be described accordingly. Since the differences between study 1 and 2 are not clearly explained, it is very confusing. It is difficult to find out the outline of research design.
  3. Methods: Under 2. Methods section, all the study methods, results, and discussions for study 1 and 2 are described. For example, 2.1.1. Pretest; 2.1.2. pretest results; 2.1.3. Main study; 2.1.4. Results; 2.1.5. Discussions. Method and Results section should be separated.
  4. Discussion: The discussions in the 2. Methods section above should move to this discussion section. Results and Discussion section can be combined. However, all the discussions should be listed under ‘Discussion’ or ‘Results and Discussion’ section.

Round 2

Reviewer 1 Report

The authors have done an excellent job responding to the reviewers' comments. In particular, most of my comments have been properly reviewed and addressed. The result is a much more readable and persuasive investigation. I think it is an improved work and the result is a much more decisive investigation.

There is only one observation to do: I understand now that the size of the anthropomorphic stickers in Figure 1 cannot be changed. However, the lack of standardization of the dimensions should be included as a limitation without distorting the quality and credibility.

Good luck with your research in the future.